# Strain *Streptomyces* sp. P-56 Produces Nonactin and Possesses Insecticidal, Acaricidal, Antimicrobial and Plant Growth-Promoting Traits

**DOI:** 10.3390/microorganisms11030764

**Published:** 2023-03-16

**Authors:** Irina Boykova, Oleg Yuzikhin, Irina Novikova, Pavel Ulianich, Igor Eliseev, Alexander Shaposhnikov, Alexander Yakimov, Andrey Belimov

**Affiliations:** 1All-Russia Institute of Plant Protection, Podbelskogo Sh. 3, Pushkin, Saint-Petersburg 196608, Russia; irina_boikova@mail.ru (I.B.);; 2All-Russia Research Institute for Agricultural Microbiology, Podbelskogo Sh. 3, Pushkin, Saint-Petersburg 196608, Russia; 3Alferov Federal State Budgetary Institution of Higher Education and Science Saint Petersburg National Research Academic University of the Russian Academy of Sciences, Khlopin Str., 8/3-A, Saint-Petersburg 194021, Russia; 4Research Center of Nanobiotechnologies, Peter the Great St Petersburg Polytechnic University, Polytechnicheskaya, 29, Saint-Petersburg 195251, Russia

**Keywords:** acaricide, insecticide, *Streptomyces*, secondary metabolites, spectral methods, nonactin, NMR, X-ray analysis

## Abstract

Streptomycetes produce a huge variety of bioactive metabolites, including antibiotics, enzyme inhibitors, pesticides and herbicides, which offer promise for applications in agriculture as plant protection and plant growth-promoting products. The aim of this report was to characterize the biological activities of strain *Streptomyces* sp. P-56, previously isolated from soil as an insecticidal bacterium. The metabolic complex was obtained from liquid culture of *Streptomyces* sp. P-56 as dried ethanol extract (DEE) and possessed insecticidal activity against vetch aphid (*Medoura viciae* Buckt.), cotton aphid *(Aphis gossypii* Glov.), green peach aphid (*Myzus persicae* Sulz.), pea aphid *(Acyrthosiphon pisum* Harr.) and crescent-marked lily aphid *(Neomyzus circumflexus* Buckt.), as well as two-spotted spider mite (*Tetranychus urticae*). Insecticidal activity was associated with production of nonactin, which was purified and identified using HPLC-MS and crystallographic techniques. Strain *Streptomyces* sp. P-56 also showed antibacterial and antifungal activity against various phytopathogenic bacteria and fungi (mostly for *Clavibacfer michiganense*, *Alternaria solani* and *Sclerotinia libertiana*), and possessed a set of plant growth-promoting traits, such as auxin production, ACC deaminase and phosphate solubilization. The possibilities for using this strain as a biopesticide producer and/or biocontrol and a plant growth-promoting microorganism are discussed.

## 1. Introduction

The recent trends in the study of bioactive secondary microbial metabolites [1,2,3] show the increasing practical importance of this field. Particularly, the metabolites produced by actinomycetes (*Actinomycetales)* are very useful agents for pharmacology and agrobiology [4,5,6,7,8]. Microorganisms of the genus *Streptomyces* are widespread actinomycetes inhabiting the soil and associating with plants [9,10]. Secondary metabolic products of actinomycetes include a huge number of chemical structures. The majority of these compounds are antibiotics (antimicrobial, antitumor or antiviral agents), but other bioactive compounds (enzyme inhibitors, pharmacologically active substances, pesticides and herbicides) were also described [11,12,13,14,15,16,17,18]. Due to numerous important properties, the bioactive metabolites of actinomycetes offer promise for applications in agriculture as plant protection products. The problem of controlling the number of harmful arthropods at an environmentally safe level is associated with their high breeding potential, high prevalence and wide polyphagia, as well as very limited range of effective biocontrol products. Various biological insecticides, including those produced by streptomycetes, have established themselves as effective means of combating harmful insects, alternatively to synthetic chemicals [5,8,10,19]. They are characterized by greater specificity and lower toxicity for landscape biota, animals and humans, as well as higher ability to degrade in natural cycles of matter. However, for the effective use of biopesticides and the solution of related problems, such as the acquisition of pest resistance, it is necessary to deeply study the mechanisms underlying the action of the products and the producing microorganisms themselves [5,16]. 

Nonactin is a biologically active compound produced by several *Streptomyces* species [20,21,22,23], and is known as the macrotetrolide antibiotic consisting of four subunits in the form of nonactic acid [24,25]. Nonactic acid and/or its derivatives (monactin, dinactin, trinactin and tetranactin) possess antibacterial, antifungal, insecticidal and acaricidal activity [11,25,26,27]. It was also shown that *Streptomyces globisporus* produced nonactine and possessed insecticidal activity against Colorado potato beetle *(Leptinotarsa decemlineata)* [20]. Growth inhibition of various pathogenic bacteria, particularly *Agrobecterium tumefaciens*, by *Streptomyces cavourensis* TN638 producing a mixture of nonactin with other derivatives of nonactic acid was described [28]. Tetranactin produced by several streptomycetes species caused growth inhibition of various bacteria and fungi, as well as acaricidal activity against two-spotted spider mite (*Tetranychus urticae*) [11]. These reports show that the role of nonactin as an insecticidal and acaricidal agent has not been studied in detail.

The biological role of streptomycetes in relation to plants is diverse, varying from pathogenesis to mutualistic symbiosis [29,30]. The significance of streptomycetes as important plant growth-promoting bacteria (PGPB) is actively debated [10,19,30,31]. In addition to biocontrol effects, the beneficial properties of streptomycetes for plants are associated with their ability to produce phytohormones auxins [32,33,34,35] and siderophores [36], to dissolve unavailable phosphates [37] and to have 1-aminocyclopropane-1-carboxylic acid (ACC) deaminase activity [38,39,40]. Streptomycetes quite often combine growth-promoting properties with antimicrobial activity, and in some cases with insecticidal [26,35] or acaricidal activity [11,26]. At the same time, they can also have negative properties for plants [29,30,31]. Therefore, for the successful application of these microorganisms, a comprehensive study of their properties is necessary.

The aim of this report was to characterize in detail the strain *Streptomyces* sp. P-56, previously isolated as an insecticidal agent (unpublished data), to identify its active insecticidal metabolite and to investigate its other beneficial or deleterious traits for plants. As a result, the possibility and prospects of using this strain as a biopreparation for plant protection and growth promotion were evaluated.

## 2. Materials and Methods

### 2.1. Microorganism

The actinomycetal strain P-56 was initially isolated from red soil originating from rice fields of India, identified as *Streptomyces loidensis* P-56 using standard morphological and biochemical tests of numerical taxonomy [9,41] and characterized by high insecticidal activity against vetch aphid (*Medoura viciae* Buck.) (unpublished data). The strain P-56 was deposited in the State Collection of Entomopathogenic and Phytopathogenic Microorganisms and their Antagonists (SCEPMA, All-Russia Institute of Plant Protection, Saint-Petersburg, Russian Federation, http://vizrspb.ru/struktura-instituta/research/mikrobiologicheskoi-zashchity-rastenii; accessed on 1 January 2020). During the experimental work, the strain P-56 was maintained in the medium M19/6 (g L^−1^): corn extract–10, soluble potato starch–10, (NH_4_)_2_SO_4_–3, NaCl–3, CaCO_3_–3, microbiological agar, 20, pH = 7.2. 

### 2.2. Identification of Strain P-56

In this study, the strain P-56 was identified by sequencing the 16S rRNA (rrs) gene using PCR primers fD1 (5′-AGAGTTTGATCCTGGCTCAG-3′) and rD1 (5′-AAGGAGGTGATCCAGCC-3′) [42]. Isolation and purification of DNA, PCR reaction and electrophoresis were performed as previously described [43]. Sequencing was performed with a 3730xl DNA Analyzer (Applied Biosystems, Waltham, MA, USA) using BigDye Terminator v3.1 Cycle Sequencing Kit (Applied Biosystems, USA) following the manufacturer’s protocol. The obtained sequence was deposited to the NCBI GenBank database under accession number OP389215.1 and was aligned using pairwise Clustal W alignment and compared with the related sequences of the available type strains using BLAST analysis. Pictures of colonies and mycelium of the strain P-56 were performed by the stereomicroscope Stemi 508 (Carl Zeiss, Jena, Germany).

### 2.3. Obtaining Laboratory Sample of the Metabolic Complex

The laboratory sample of the metabolic complex was obtained from liquid culture of *Streptomyces* sp. P-56 in the form of the dried ethanol extract (DEE). For this purpose, the strain was cultivated on a rotary shaker (230 rpm) at 28 °C for 4 days in Erlenmeyer flasks containing 150 mL of liquid medium M5 (g L^−1^): glucose–10, soy flour–10, NaCl–5, CaCO_3_–3, pH = 7.2. The culture was centrifuged at room temperature for 15 min at 5000× *g* and the separated mycelium was extracted with ethanol for 1 h at constant stirring at 1500 rpm. The organic extract was separated from the mycelium by centrifugation for 20 min at 5000× *g* and 20 °C and concentrated on vacuum evaporator RE-501 (Lanphan, Zhengzhou, China) at 45 ÷ 55 °C and 0.03 ÷ 0.005 MPa up to 10 mL. The DEE concentrate was dried at 35 °C. As a result, a light brown thick paste having moisture of about 80% was obtained and stored at 8 °C for further experiments.

### 2.4. Insecticidal Activity of DEE

The contact insecticidal activity in vitro of *Streptomyces* sp. P-56 was determined under laboratory conditions against vetch aphid (*Medoura viciae* Buckt.), cotton aphid *(Aphis gossypii* Glov.), green peach aphid (*Myzus persicae* Sulz.), pea aphid *(Acyrthosiphon pisum* Harr.) and crescent-marked lily aphid *(Neomyzus circumflexus* Buckt.). Populations of harmful arthropods were bred and maintained under laboratory conditions using common bean (*Phaseolus vulgaris* L.) variety Russian Black plants. Filter paper was placed into Petri dishes (40 mm in diameter) and moisturized with 3 mL of water diluted DEE in a range of concentrations from 0.5 to 10 g L^−1^. Test insects (20 individuals of each insect per replicate, 5 replicates per DEE concentration) were placed into the dishes and incubated at 24 °C. The percentage of dead insects was estimated relative to untreated controls after 2, 4, 6 and 24 h. The results were subjected to a probit analysis to determine the LC_50_, LC_90_ and LC_98_ values of DEE toxicity.

The experiment to assess the insecticidal activity in vivo against green peach aphid was carried out using bell pepper (*Capsicum annuum* L.) cultivar Lastochka. For this purpose, young plants were infected with peach aphid and cultivated in a greenhouse under controlled environment conditions. Immediately after the number of aphids on each plant were counted, the plants were treated with water suspension of 2 g L^−1^ DEE or with 1 g L^−1^ of Confidor^®^ Extra (Bater AG, Leverkusen, Germany) containing imidacloprid (CAS138261-41-3) using a laboratory sprayer Marolex Profession 5 (Dziekanów Leśny, Marolex, Poland). Control plants were treated with tap water. The counting of live insects on each plant (4 plants per treatment) was carried out after 24 h. 

### 2.5. Isolation and Purification of the Toxin

Preparative chromatographic separation of DEE was performed on silica gel (Merck 60) using MPLC system Buchi Sepacore (Büchi Labortechnik AG, Flawil, Switzerland) completed UV-Monitor C-630, two Pump Module C-605, Control Unit S-620, fractions collector C-660. The solvents were (A) n-hexane, (B) EtOAc and (C) MeOH. About 150 mg of DEE was dissolved in 10 mL of methanol and sorbed into 3 g of silica gel. The resulting material was introduced into the upper part of the Glass Column Buchi 15/230, filled with the same silica gel and preconditioned in the flow of hexane (15 mL/min for 5 min). Chromatography conditions: flow rate 15 mL/min; detection was carried out at a wavelength of 254 nm. 

Purity and the composition of fractions were estimated by UPLC method using the Acquity H-class chromatograph (Waters Corporation, Milford, MA, USA) with PDA detector and Xevo TQD MS-detector. Then insecticidal activity in vitro of the compound 1 was verified using vetch aphid as described above.

### 2.6. Identification of the Toxin 

Melting point determination was performed on a PTP-M apparatus (Khimlabpribor, Klin, Russia). The infrared (IR) spectrum was recorded in CHCl_3_ on a Specord 75IR spectrometer (Analytik Jena, Jena, Germany) and the ultraviolet (UV) spectrum was recorded in EtOH on a Beckman Coulter DU 800 spectrometer (Beckman Coulter Inc., Brea, CA, USA). ^1^H and ^13^C nuclear magnetic resonance (NMR) spectra were recorded on a Varian DirectDrive NMR System (Varian, Palo Aho, CA, USA) in CDCl_3_ at 700 and/or 175.8 MHz. CDCl_3_ was used as an internal standard. Mass spectrometry with electrospray ionization MS (ESI) was performed using LC-MS high-resolution spectrometer MaXis (ESI-QTOF) (Bruker Corporation, Billerica, MA, USA), with direct injections of sample solutions into the ionization chamber. X-ray diffraction was measured on Kappa Apex II device (Bruker Corporation, Billerica, MA, USA) by use of CuKα radiation (λ = 1.54178 Å). Intensity of reflexes was integrated and scaled by means of the Saint and Sadabs programs from an Apex II package (Bruker AXS Inc., Madison, WI, USA). All subsequent procedures for the decision and specification of the compound structure were held in the software package of Olex2 (https://www.olexsys.org/olex2/, accessed on 1 January 2020) [44]. 

### 2.7. Method of Growing Crystals for X-ray Analysis 

To obtain crystals for X-ray diffraction, the obtained nonactin was dissolved in acetonitrile at a concentration of 4 mg/mL at a temperature of 45 °C; next, the solution was slowly cooled down to 4 °C in a Thermos bottle. Elongated crystals of 0.1–0.2 mm were usually formed on day 2–3. For X-ray diffraction, a single 0.02 × 0.02 × 0.2 mm crystal was extracted using a special loop, mounted on a goniometer, and frozen in a 100 K nitrogen-cooled stream.

### 2.8. Acaricidal Activity of DEE

Fresh leaves of common bean variety Russian Black were placed on cotton rafts in pallets with water, and each leaf was infected with 10 two-spotted spider mites (*Tetranychus urticae*). Then, leaves were sprayed with water suspensions of DEE in concentrations ranging from 1.25 to 10 g L^−1^ using a laboratory sprayer Mercury PRO+ (Kwazar Corporation, Budy Grzybek, Poland). Control leaves were treated with tap water. An aqueous solution of the commercial biopreparation Fitoverm (Farmbiomed, Russia) in a concentration 2 g L^−1^ of active substance Aversectin-C (CAS73989-17-0) was used as a positive control according to manufacturer’s instructions. Six leaves were prepared for each treatment located on individual cotton rafts. After the rafts were incubated in a greenhouse for 24 h, the number of alive spider mites was counted on each leaf. 

### 2.9. Antifungal and Antibacterial Activity of Strain P-56

The antagonistic properties of *Streptomyces* sp. P-56 were determined by the well diffusion method as previously described [17]. Twenty test microorganisms, including various species of bacteria and fungi obtained from the SCEPMA, were used, and are described in Table 1. The diameter of the growth inhibition zones of the test microorganisms induced by *Streptomyces* sp. P-56 was measured after 5 days of inoculation at 28 °C. 

### 2.10. Acetylene Reduction Activity of Strain P-56

Acetylene reduction (nitrogen-fixing) activity of *Streptomyces* sp. P-56 was determined using the acetylene reduction assay [45]. Briefly, the strain was cultivated in flasks (*n* = 3) containing 50 mL liquid MDF medium for 5 days at 24 °C. The composition of MDF medium was as follows (g L^−1^): glucose, 1; sucrose, 1; gluconic acid, 1; citric acid, 1; malic acid, 1; mannitol, 1; starch, 0.5; KH_2_PO_4_, 2; Na_2_HPO_4_, 3; MgSO_4_, 0.2; NaCl, 0.1; pH 6.8. Then, the flasks were closed with rubber stoppers, and 10% of the gas phase was replaced with acetylene and incubated for 1 day. Reduction of acetylene to ethylene was measured using gas chromatograph GC-2014 (Shimadzu Corporation, Kyoto, Japan).

### 2.11. Phytohormone Production by Strain P-56

Strain *Streptomyces* sp. P-56 was grown in vials with 5 mL of Chapek (three replicates) liquid medium amended with 1% L-tryptophan for 5 days at 24 °C. To determine phytohormones (auxins, abscisic acid, salicylic acid, gibberellic acid GA3), the culture was centrifuged at 11,000× *g* for 15 min and the supernatants were acidified to pH 3.0 with 0.4 N hydrochloric acid and extracted with equal volumes of ethyl acetate. The organic phase containing phytohormones was evaporated to dryness under vacuum at 35 °C and suspended in 0.5 mL of 18% acetonitrile. All obtained samples were filtered through 0.22 μm Costar^®^ Spin-X^®^ microtubes with nylon membrane filters (Sigma-Aldrich Int. GMBH) prior to chromatographic analysis. Uninoculated medium was used as a control. Phytohormones were separated on Waters ACQUITY UPLC BEH RP18 Shield (1.7 µm, 2.1 × 50 mm) column (Waters, Milford, MA, USA) in mixture of 0.1% formic acid (A) and acetonitrile supplied with 0.1% formic acid (B) at flow rate 0.3 mL min-1 with isocratic elution in 18% B for 5 min, followed by washing with 80% B for 2 min and conditioning the column for 3 min at 18% B. Auxins and salicylic acid were detected with fluorescence detector (ƛex = 280 nm, ƛem = 350 nm for auxins; ƛex = 300 nm, ƛem = 405 nm for salicylic acid). Gibberellic acid GA3 and abscisic acid were detected with Waters eλPDA Detector at wavelengths of 208 and 265 nm, respectively. 

### 2.12. Utilization of ACC by Strain P-56

Strain *Streptomyces* sp. P-56 was cultivated on a liquid salt minimal (SM) medium [46] containing no nitrogen source (negative control treatment) and supplemented with 250 mg L^−1^ NH_4_NO_3_ (positive control treatment) or with 250 mg L^−1^ ACC. Bacteria were cultivated using shaking at 200 rpm for 5 days at 28 °C. Then, the obtained suspensions (2 replicates per treatment) were intensively vortexed, and optical density (OD) was measured at 600 nm using spectrophotometer SmartSpec Plus (Bio-RAD, Hercules, CA, USA). 

### 2.13. Solubilization of Phosphates by Strain P-56

Phosphate solubilization activity was determined as described previously [47]. Strain *Streptomyces* sp. P-56 was cultivated on modified Pikovskaya (PKV) agar medium containing (g L^−1^): glucose, 5; fructose, 5; yeast extract, 0.5; (NH_4_)_2_SO_4_, 0.5; MgSO_4_·7H_2_O, 0.1; KCl, 0.2; NaCl, 0.2; MnSO_4_·4H_2_O, 0.02; FeSO_4_·7H_2_O, 0.02; agar, 15; pH 7.0. The PKV medium was supplemented with P source (5 g L^−1^): Ca_3_(PO_4_)_2_, Ca-phytate), AlPO_4_ or FePO_4_. The plates were spot-inoculated (3 replicates per P source) and incubated at 25 °C for 7 days. Diameter of visible halo zones around colonies was measured. 

### 2.14. Production of Siderophores by Strain P-56

Siderophore production was determined using a chrome azurol S (CAS) shuttle solution as previously described [48]. The assay was calibrated by generating a standard curve for samples containing 1 to 100 µM deferoxamine mesylate (DFM). 

### 2.15. Bioassay with Plants

Strain *Streptomyces* sp. P-56 was cultivated on a liquid medium M5 for 5 days at 28 °C, supernatant was obtained after centrifugation of the culture for 15 min at 5000× *g* and mycelium was suspended in sterile tap water with a final concentration of 10^7^ colony forming units per mL. Seeds of barley (*Hordeum vulgare* L.) variety Samson and watercress (*Nasturtium officinale* W.T.Aiton) variety Zabava were surface-sterilized by treatment with 5% sodium hypochlorite for 15 min. One part of seeds was placed into vials containing sterile tap water (control treatment), supernatant of *Streptomyces* sp. P-56 or solution of 2 g L^−1^ DEE and soaked for 2 h at 22 °C. Then, the seeds were transferred to Petri dishes (four dishes with 20 seeds per treatment) with wet filter paper. Another part of seeds was transferred to Petri dishes (4 dishes with 20 seeds per treatment) with filter paper wetted with sterile tap water or suspension of mycelium. All Petri dishes were incubated for 5 days at 22 °C. Then, the number of germinated seeds, the main root length and the root number on barley were determined. 

### 2.16. Statistical Analysis

Statistical analysis of the data was performed using the software STATISTICA version 10 (TIBCO Software Inc., Palo Alto, CA, USA). MANOVA analysis with Fisher’s LSD test, Student’s *t*-test and Probit analysis were used to evaluate differences between means and to estimate significance of the effects.

## 3. Results

### 3.1. Identification of Strain P-56

The sequence of the strain P-56 showed very high percent identity (PI) with sequences of several types of strain belonging to different species of the genus *Streptomyces*, namely the *St. arboris* strain TRM68085 (PI = 100.00%; NZ_VYUA01000026.1), *St. cyaneofuscatus* strain NRRL B-2570 (PI = 99.93%; NZ_JOEM01000050.1), *St. fulvorobeus* strain DSM 41455 (PI = 99.93%; NZ_JACCCF010000001.1), *St. albovinaceus* strain NRRL B-2566 (99.86%; NZ_MUAX01000176.1), *St. rubiginosohelvolus* strain JCM 4415 (PI = 99.86%; NZ_BMTW01000038.1), *St. griseolus* strain NRRL B-2925 (PI = 99.79%; NZ_JOFC01000069.1) and *St. californicus* strain FDAARGOS (PI = 99.79%; NZ_CP070242.1). Therefore, the studied microorganism was assigned as *Streptomyces* sp. strain P-56. Characteristic pictures of colonies and mycelium of the strain Streptomyces sp. P-56 are present in Figure 1. 

### 3.2. Insecticidal Activity 

Under in vitro conditions, treatment of various aphid species with DEE caused mortality from 60% to 100%, depending on the test insects and concentration of DEE (Table 2). The most sensitive to the treatment with DEE was *A. gossypii*, whereas *A. pisum* showed a maximum resistance relative to other species (Table 2). The death of aphids was accompanied by the following pathological symptoms: difficulties in moving, paling and yellowing of the cuticle, slight swelling of the body, more brittle and brittle limbs and complete paralysis in the end of experiment. The insect covers had acquired a yellowish hue; the abdomen was swollen and acquired an oily sheen. The broken limbs were observed, suggesting the increase in their fragility. 

Treatment of peach aphids on the infected bell pepper leaves in vivo caused 89% mortality of this pest within 24 h (Figure 2). Insecticidal activity of DEE was comparable with the effect of Confidor^®^ Extra causing 98% mortality of peach aphids.

### 3.3. Isolation and Structural Identification of the Active Substance

The obtained DEE from *Streptomyces* sp. P-56 mycelium was subjected to chromatographic separation by MPLC, as shown in the Materials and Methods section. The gradient chromatographic mode and procedure of selection of the fractions are shown in Table 3. 

The obtained fractions were tested for insecticidal activity in vitro against vetch aphid, as described above. Fractions 4÷8 showed insecticidal activity with a maximum activity of fraction 5. HPLC-MS studies showed that all active fractions contained the same major component with a molecular weight of 736. Recrystallization of the obtained fractions from ethanol gave 49.5 mg of white crystalline substance, which was identified as compound 1.

As a result, a white crystalline compound 1 was obtained and showed high toxicity against vetch aphid (mortality rate 92%) after treatment with 1 mg L^−1^ water solution for 2 h. The results of the chemical structure identification of compound 1 obtained by NMR methods, including both one-dimensional ^1^H (Figure 3A) and ^13^C (Figure 3B), as well as two-dimensional (HSQC, HMBC, C2H2QC, TOCSY) experiments, indicated the structure of a lactone nature. 

However, based on the data of high-resolution mass spectroscopy, compound 1 should have a molecular weight of 737.44, which corresponds to its tetramer. The spectral card of compound 1 was as follows: (1R,2R,5R,7R,10S,11S,14S,16S,19R,20R,23R,25R,28S,29S,32S,34S)-2,5,11,14,20,23,29,32-Octamethyl-4,13,22,31,37,38,39,40-octaoxapentacyclo [32.2.1.1^7,10^.1^16,19^.1^25,28^]tetracontane-3,12,21,30-tetrone: mp: 144–146 °C.^1^H NMR: 4.96 (4H, ddq, J = 7.8, 6.2, 5.6 Hz, H-8), 4.00 (4H, ddd, J = 7.5, 7.2, 7.2 Hz, H-3), 3.84 (4H, ddd, J = 13.6, 7.5, 5.9 Hz, H-6), 2.49 (4H, dq, J = 7.1, 7.2 Hz, H-2), 1.98 (4H, m, H-5a), 1.92 (4H, m, H-4a), 1.77 (4H, ddd, J = 13.7, 7.8, 5.9 Hz H-7a), 1.72 (4H, ddd, J = 13.7, 7.5, 5.6 Hz, H-7b), 1.61 (4H, m, H-4b), 1.48 (4H, m, H-5b), 1.22 (12H, d, J = 6.2, H-9), 1.08 (12H, d, J = 7.1, H-10). ^13^C NMR: 176.9 (C-1, C(O) = O), 82.7 (C-3, C–O), 79.0 (C-6, C–O), 71.7 (C-8, C–O), 47.9 (C-2), 44.9 (C-7), 34.0 (C-5), 30.8 (C-4), 23.1 (C-9, CH_3_), 15.5 (C-10, CH_3_). ESI–TOF MS m/z (relative intensity): 738.4505 (C_40_H_65_O_12_, calcd 738.4512) [M + H + 1]^+^ (30), 737.4474 (C_40_H_65_O_12_, calcd 737.4479) [M + H]^+^ (60). Based on the obtained spectral characteristics and the data available in the literature [49,50], we assumed that the isolated compound 1 is nonactin.

Nonactin has 16 chiral centers; therefore, it was necessary to confirm the configuration of the molecule of the isolated compound 1 for completing structural identification. Summarized results of X-ray data acquisition and processing are given in Table 4. As a result, the structure of nonactin was obtained (Figure 4) and confirmed that compound 1 is nonactin.

Crystallographic data (excluding structure factors) for the obtained structures were deposited to the Cambridge Crystallographic Data Centre (CCDC) as supplementary publication No 1893406 CCDC. A copy of the data can be obtained on application to the CCDC (deposit@ccdc.cam.ac.uk).

Insecticidal activity in vitro of the obtained nonactin was verified using vetch aphid. The nonactin was very toxic for vetch aphids (Figure 5) and its insecticidal activity was comparable with DEE (Table 2, Figure 2).

### 3.4. Acaricidal Activity

Treatment with 1.25 g L^−1^ DDE was toxic for two-spotted spider mites, resulting in 56% mortality, whereas treatment with 2.5 g L^−1^ DDE caused almost compete mortality of this pest, and the effect was comparable with commercial Fitoverm (2.0 g Aversectin-C L^−1^) treatment (Table 5). DEE induced the following pathological symptoms: partial and then complete paralysis of the spider mites, resulting in their death. The color of the cuticle of the dead ticks had a darker shade. The dead individuals were mummified, apparently due to the loss of moisture through the integument.

### 3.5. Antagonistic Properties

The strain *Streptomyces* sp. P-56 inhibited the growth of 5 out of 8 bacterial test strains and 8 out of 12 fungal strains (Table 1). Maximal diameter of the growth-inhibiting zone was observed for the bacterium *Clavibacfer michiganense* subsp. *michiganense* VIZR-13a as well as for the fungi *Alternaria solani* VIZR-32 and *Sclerotinia libertiana* VIZR-32-1.

### 3.6. Plant Growth-Promoting Traits

The strain *Streptomyces* sp. P-56 produced the phytohormones indolil-3-acetic (24 ± 4 µg mL^−1^), indolyl-3-carboxylic (46 ± 1 µg mL^−1^), indolyl-3-lactic (28 ± 3 µg mL^−1^) and salicylic (14 ± 5 µg mL^−1^) acids, but did not produce abscisic and gibberellic acids. No acetylene reduction (nitrogen fixation) activity or endogenous ethylene production was detected in the culture of *Streptomyces* sp. P-56. Active growth of *Streptomyces* sp. P-56 was observed in a selective SM medium containing ACC as a sole nitrogen source (OD_600_ = 1.3 ± 0.1), and its growth was similar in the presence of mineral nitrogen (OD_600_ = 1.1 ± 0.1). This suggested that the strain possesses ACC deaminase activity. A halo zone around the colony of *Streptomyces* sp. P-56 was detected only in the presence of Ca-phytate (diameter 4 ± 0.5 mm). However, this strain did not produce siderophores.

### 3.7. Biotests with Plants

Treatments of barley and watercress seeds with supernatant of *Streptomyces* sp. P-56 or DEE had no effect on the seed germination of barley and watercress (Table 6). Control seedlings of barley had longer roots by 18% compared to those treated with the supernatant, but the root length of watercress was not affected by either treatment. The seed germination and root lengths of either plant species were not affected by DEE treatment (Table 6).

Decreased root and shoot elongation was observed when watercress seeds were inoculated with live mycelium of *Streptomyces* sp. P-56 (Figure 6A). However, inoculation had no effect on the elongation of barley roots and shoots (Figure 6B), as well as on the root number. Negative effects on roots of both plant species, such as discoloration, necrosis and other lesions, were not detected.

## 4. Discussion

Antibiotic nonactin was firstly isolated from mycelium of *Streptomyces viridochromogenes* [51]. Until 1991, nonactin had been found in six more species of the genus *Streptomuces* [20]. In recent years, this list has expanded due to the species *St. cavourensis* [28] and several unidentified *Streptomyces* spp. [21,22]. Growth inhibition of pathogenic bacteria *Agrobecterium tumefaciens*, *Salmonella typhimurium* and *Staphylococcus aureus* by nonactin-producing *St. cavourensis* TN638 was described [28]. Antimicrobial effects of nonactic acid produced by several strains of Streptomyces were also reported [11,26,27]. Here, we demonstrate that nonactin-producing *Streptomyces* sp. P-56 possesses antagonistic activity against *Bacillus subtilis, Clavibacfer michiganense, Erwinia caratovora, Streptomyces globisporus, Xanthomonas campestris* and a number of phytopathogenic fungi belonging to various taxonomic groups (Table 1).

Information about insecticidal activity of streptomycetes via nonactin production is very limited. Namely, *St. globisporus* strain 0234 showed insecticidal activity against Colorado potato beetle [20]. Nonactic acid (precursor of nonactin) was also highly toxic for Colorado potato beetle, Mexican bean beetle (*Epilachna varivestis*) and brown-tail moth (*Euproctis chrysorhoea*) [26]. Our results made it possible to substantially fill this gap, demonstrating the toxicity of DEE obtained from *Streptomyces* sp. P-56 against five aphid species, and proving that nonactin is an active compound causing the death of these insects (Table 2). Previously, acaricidal activity (LC_50_ = 2 mg mL^−1^) was observed after treatment of *Tetranychus cinnabarinus*, *T. kanzawa* and *T. urticae* with a mixture of nonactic and homononactic acids [26]. Acaricidal effects of tetranactin against *T. urticae* and *Tyrophagus dimidiatus* were also described [11]. This is the first report demonstrating acaricidal activity against *T. urticae* of nonactin-producing streptomycetes, namely the strain *Streptomyces* sp. P-56.

Macrotetralids, including nonactin, are membrane-active compounds that violate the permeability of the membranes, particularly blocking the activity of potassium channels and inducing disturbance to the transport of ions into cells [11,25,52]. As a consequence, violation of oxidative phosphorylation, ATP hydrolysis, swelling of mitochondria, lysis of zoospores and other toxic effects were observed [11,53]. These mechanisms of action of macrotetralids may be a partial explanation for the wide spectrum of action on various organisms, including prokaryotic fungi, insects and mites.

Obtaining high-quality nonactin crystals and their X-ray diffraction analysis are associated with certain difficulties, which were previously described in detail [54]. In particular, nonactin crystals without metal ions have a primitive monoclinic cell with twinning. At that, the asymmetric part of the crystal cell contains two unbounded halves of the nonactin molecule, slightly different in conformation. Here, we used an original method of growing nonactin crystals (see Materials and Methods section). The problem for determination of the structure of nonactin molecule in the crystallographic group P2/*n* was solved using the SHELXT [55] software and clarification of its structure was carried out taking into account the twinning [56]. As a result, a high-quality nonactin crystal was obtained without the use of metal ions, and its X-ray diffraction analysis was performed.

Streptomycetes are well-known plant-associated microorganisms having a number of contradictory (phytopathogenic and plant growth-promoting) properties [10,19,29,30,31]. Therefore, it was important to investigate the traits of *Streptomyces* sp. P-56 related to plant–microbe interactions. Apart from antagonistic activity against phytopathogenic bacteria and fungi, this strain possessed beneficial properties, such as the production of auxins and the utilization of ACC and solubilization of Ca-phytate. Our results confirmed that rhizosphere bacteria of the genus *Streptomyces* could be considered as PGPB. Indeed, auxin production [32,33,34,35], ACC deaminase activity [38,39,40,57] and phosphate solubilization (Jog-2014) were described for various plant-associated streptomycetes. The combination of plant growth-promoting traits with antimicrobial [37], insecticidal [26,35] or acaricidal activity [11,26] was also reported. At the same time, the strain *Streptomyces* sp. P-56 did not promote plant growth in the bioassay with watercress and barley, and even DEE treatment reduced the root elongation of barley (Table 6), and leaf cells reduced the elongation of watercress seedlings (Figure 6). We expected increased root elongation induced by leaf cells of *Streptomyces* sp. P-56, since this is a biotest for the evaluation of growth-promoting activity by bacteria containing ACC deaminase [58,59]. It is possible that such a plant response was associated with some unknown compounds produced by *Streptomyces* sp. P-56 and inhibiting elongation. It is known that some phytopathogenic *Streptomyces* species synthesize phytotoxins [60]. Interestingly, decreased root elongation of watercress was observed only when the plants were inoculated with live mycelium, but was absent after treatment with DEE. The results suggested that production of the growth-inhibiting compound was induced by the plant. On the other hand, *St. griseus* produced nonactic acid and stimulated elongation of cucumber shoot, whereas root growth was unaffected [26]. It is likely that the plant’s response to inoculation with streptomycetes depends on the concentration of substances produced by them and the activity of bacterial enzymes under the conditions of plant–microbe association. More detailed study is required to understand the effect of *Streptomyces* sp. P-56 on plant growth and to evaluate the possibility for using this strain as a biopesticide producer and/or a biocontrol microorganism.

## 5. Conclusions

The strain *Streptomyces* sp. P-56 has a wide range of biological activity, which is expressed in the growth inhibition of various phytopathogenic bacteria and fungi, high toxicity to various aphid species and also in the manifestation of acaricidal activity against two-spotted spider mites. Insecticidal activity of this strain was associated with the production of nonactin, which was found in the metabolic complex, extracted with DEE, purified and identified using HPLC-MS and crystallographic techniques. Antimicrobial and acaricidal activities of *Streptomyces* sp. P-56 could be also associated with nonactin; however, more detailed study is needed to prove this hypothesis. It should be noted that information about toxic effects of pure nonactin on microorganisms, insects, mites and plants is limited, and this makes it difficult to interpret the results. The obtained results of bioassay with plants suggest that this strain can produce unknown growth-inhibiting substances, particularly when bacteria interact with plants. At the same time, *Streptomyces* sp. P-56 possesses a set of plant growth-promoting traits, such as the inhibition of phytopathogens, production of auxins, utilization of ACC and solubilization of organic phosphates. These properties can probably be manifested when plants are inoculated with this strain under natural conditions, as a result of which a biocontrol (against microorganisms, insects, and mites) and growth-stimulating effect can be expected. To verify this speculation, it is necessary to carry out vegetative and field experiments.

## Figures and Tables

**Figure 1 microorganisms-11-00764-f001:**
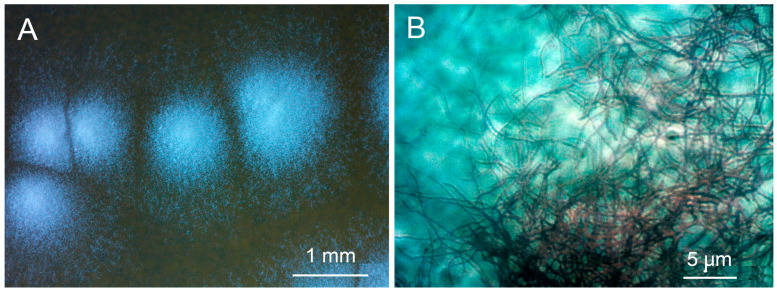
Appearance of the colonies (**A**) and mycelium (**B**) of the strain *Streptomyces* sp. P-56.

**Figure 2 microorganisms-11-00764-f002:**
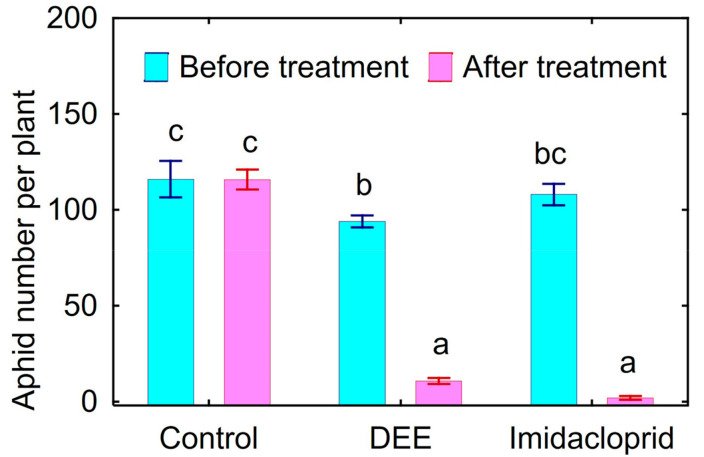
Toxicity of the dried ethanol extract (DEE) of *Streptomyces* sp. P-56 and imidacloprid against peach aphid in greenhouse experiment with bell pepper cultivar Lastochka in vivo. Vertical bars show standard error. Different letters show significant differences between treatments (least significant difference test, *p* < 0.05, *n* = 4).

**Figure 3 microorganisms-11-00764-f003:**
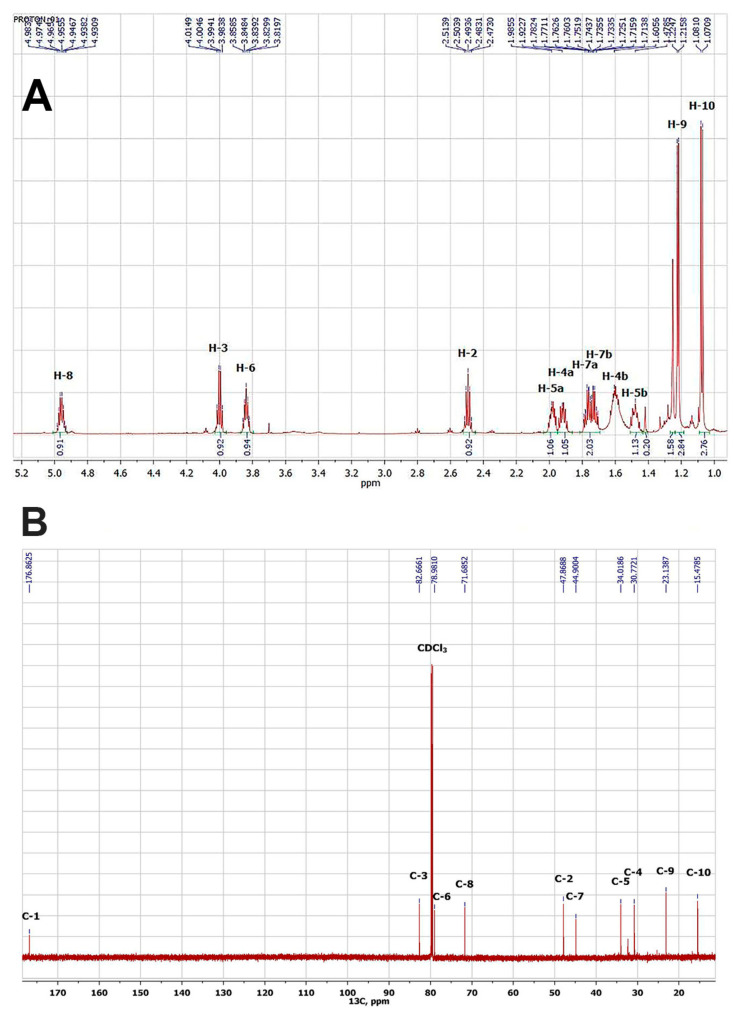
Spectra NMR ^1^H of compound 1, CDCl_3_, 700 MHz (**A**) and NMR ^13^C of compound 1, CDCl_3_, 175.8 MHz (**B**).

**Figure 4 microorganisms-11-00764-f004:**
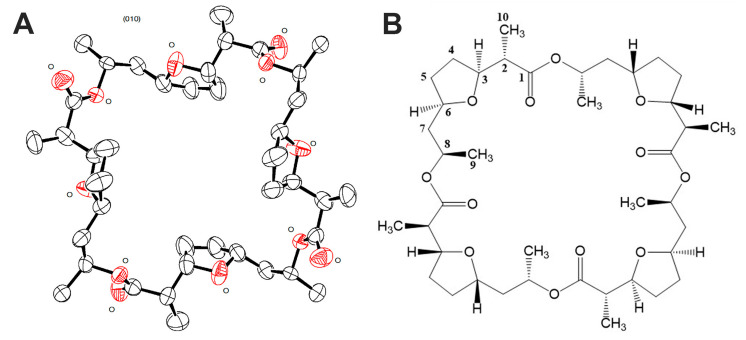
Structure of nonactin obtained as a result of X-ray diffraction analysis (**A**) with view perpendicular to the crystal plane (010). The asymmetric part of the unit cell contains two unbound halves of the molecule. Thermal ellipsoids correspond to 50% probability level. Oxygen atoms are marked with O. Structure formula of nonactin (**B**).

**Figure 5 microorganisms-11-00764-f005:**
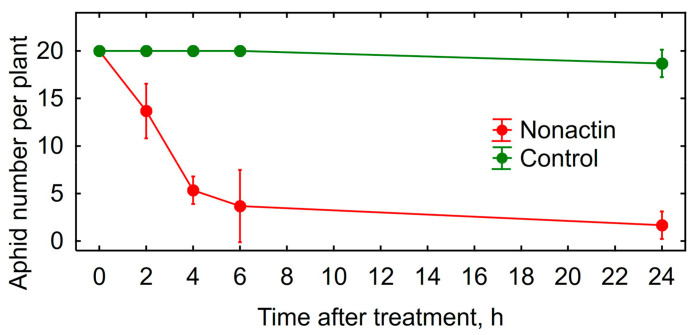
Toxicity of nonactin against vetch aphid in greenhouse experiment with bell pepper in vivo. Vertical bars show confidence intervals (*p* = 0.05, *n* = 4).

**Figure 6 microorganisms-11-00764-f006:**
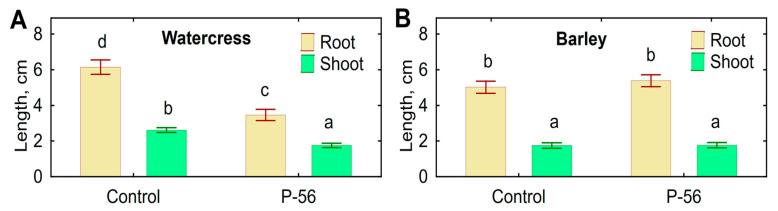
Effect of leaf cells of *Streptomyces* sp. P-56 on root (**A**) and shoot (**B**) length of barley and watercress seedlings. Vertical bars show standard error. Different letters show significant differences between treatments (least significant difference test (*p* < 0.05, *n* = 70).

**Table 1 microorganisms-11-00764-t001:** Antagonistic activity of *Streptomyces* sp. P-56 against bacteria and fungi.

Microorganism	Strain Name	Diameter of the Growth Inhibition Zone, mm
**Bacteria**		
*Bacillus subtilis*	VIZR-B10	13 ± 1 ab
*Clavibacfer michiganense* subsp. *michiganense*	VIZR-13a	35 ± 3 d
*Erwinia caratovora* subsp. *caratovora*	VIZR: 160	22 ± 3 c
*Escherichia coli*	T-047	0
*Pseudomonas fluoresceus*	VKM-B-894	0
*Staphylococcus aureus*	209p	12 ± 1 ab
*Streptomyces globisporus*	VKM-Ac-179	10 ± 1 a
*Xanthomonas campestris* pv. *vesicatoria*	VIZR-322	0
**Fungi**		
*Alternaria solani*	VIZR-32	24 ± 3 c
*Ascochyta melonis*	VIZR-2-21	0
*Aspergillus niger*	ATCC16404	0
*Fusarium graminearum*	COP	12 ± 1 ab
*Fusarium redolens*	VIZR-11	15 ± 1 b
*Fusarium sambucinum*	VIZR-18	12 ± 1 ab
*Fusarium solani*	VIZR-38	20 ± 3 c
*Penicillium granulatum*	ATCC-10450	0
*Rhizoctonia solani*	VIZR-28	15 ± 2 b
*Saccharomyces cerevisiae*	K-12	0
*Sclerotinia libertiana*	VIZR-32-1	25 ± 3 c
*Trichoderma gypseum*	VIZR-G-24	10 ± 2 a

Data are mean ± standard error. Different letters show significant differences between treatments (least significant difference test, *p* < 0.05, *n* = 3).

**Table 2 microorganisms-11-00764-t002:** Toxicity of the dried ethanol extract (DEE) of *Streptomyces* sp. P-56 against different aphid species in laboratory experiments in vitro.

Aphid Species	Lethal Concentrations, mg mL^−1^		
	LC_50_	LC_90_	LC_98_
*Medoura viciae*	1.1	4.2	6.0
*Aphis gossypii*	0.6	3.2	4.7
*Myzus persicae*	0.3	3.5	5.4
*Acyrthosiphon pisum*	1.8	6.9	10.0
*Neomyzus. circumtlexus*	2.6	5.0	6.5

**Table 3 microorganisms-11-00764-t003:** Chromatographic mode and procedure for selection of the fractions having insecticidal activity against *Medoura viciae* Buckt.

Fraction Collection Time	Starting Solvents Ratio, Vol %	Final Solvents Ratio, Vol %	Fraction Number	Fraction Volume, mL	Aphid Mortality, %
0 min–2 min 55 s	A 100	A 100	1	45	37
2 min 55 s–3 min	A 100	A–B 90:10			
3 min–5 min 55 s	A–B 90:10	A–B 90:10	2	45	42
5 min 55 s–6 min	A–B 90:10	A–B 80:20			
6 min–8 min 55 s	A–B 80:20	A–B 80:20	3	45	40
8 min 55 s–9 min	A–B 80:20	A–B 70:30			
9 min–11 min 55 s	A–B 70:30	A–B 70:30	4	45	93
11 min 55 s–12 min	A–B 70:30	A–B 60:40			
12 min–14 min 55 s	A–B 60:40	A–B 60:40	5	45	100
14 min 55 s–15 min	A–B 60:40	A–B 50:50			
15 min–17 min 55 s	A–B 50:50	A–B 50:50	6	45	92
17 min 55 s–18 min	A–B 50:50	A–B 40:60			
18 min–20 min 55 s	A–B 40:60	A–B 40:60	7	45	87
20 min 55 s–21 min	A–B 40:60	A–B 30:70			
21 min–23 min 55 s	A–B 30:70	A–B 30:70	8	45	57
23 min 55 s–24 min	A–B 30:70	A–B 20:80			
24 min–26 min 55 s	A–B 20:80	A–B 20:80	9	45	42
26 min 55 s–27 min	A–B 20:80	A–B 10:90			
27 min–29 min 55 s	A–B 10:90	A–B 10:90	10	45	40
29 min 55 s–30 min	A–B 10:90	B 100			
30 min–32 min 55 s	B 100	B 100	11	45	32
32 min 55 s–33 min	B 100	B 100			
33 min–36 min	C 100	C 100	12	45	34
36 min–39 min	C 100	C 100	13	45	0

The solvents were (A) *n*-hexane, (B) EtOAc and (C) MeOH.

**Table 4 microorganisms-11-00764-t004:** X-ray data acquisition and processing results.

Formula	C_40_H_64_O_12_
Molar mass	736.91
Diffractometer	Kappa Apex II (Bruker AXS)
Temperature (K)	100.0
Radiation source	CuKα (λ = 1.54178 Å)
Space group	P2/*n*
Cell parameters: a, b, c (E)	28.5271(9), 5.6924(2), 28.5193(9)
б, в, г (°)	90, 113.044(2), 90
Volume (E^3^)	4261.6(2)
Z	4
The range of angles 2I (°)	3.366–105.908
Resolution (E)	26.2508–0.9658
Index range	−29 ≤ h ≤ 29, −5 ≤ k ≤ 5, −29 ≤ l ≤ 28
Total reflexes	68422
Independent reflexes	4887
Completeness (%)	98.97
Average I/y(I)	52.64
R_int_	0.0449
R_sigma_	0.0190
Data/parameters	4887/478
R [I > =2y (I)]	R_1_ = 0.1067, wR_2_ = 0.2448

**Table 5 microorganisms-11-00764-t005:** Acaricidal activity of the dried ethanol extract (DEE) of *Streptomyces* sp. P-56 against two-spotted spider mite (*Tetranychus urticae*) on the infected common bean leaves.

Treatments	*Tetranychus urticae* Mortality, %
Tap water	Nd
1.25 g DEE L^−1^	56 ± 8 a
2.5 g DEE L^−1^	89 ± 5 b
5.0 g DEE L^−1^	98 ± 4 b
10.0 g DEE L^−1^	100 ± 3 b
20.0 g DEE L^−1^	95 ± 5 b
Fitoverm (2.0 g Aversectin-C L^−1^)	95 ± 5 b

Data are mean ± standard error. Nd means that no dead mites were found. Different letters show significant differences between treatments (least significant difference test, *p* < 0.05, *n* = 6).

**Table 6 microorganisms-11-00764-t006:** Effect of culture supernatant and dry ethanol extract (DEE) of *Streptomyces* sp. P-56 on seed germination and root length of barley and watercress seedlings.

Treatments	Seed Germination, %		Root Length, mm	
	Barley	Watercress	Barley	Watercress
Control	58 ± 5 a	94 ± 2 b	108 ± 5 b	33 ± 2 a
Supernatant	73 ± 6 a	85 ± 5 b	89 ± 4 a	31 ± 3 a
2 g L^−1^ DEE	45 ± 6 a	85 ± 3 b	105 ± 5 b	37 ± 2 a

Data are mean ± standard error. Different letters show significant differences between treatments (least significant difference test, *p* < 0.05, *n* = 4 for seed germination; *n* > 35 for root length). DEE stands for dry ethanol extract of *Streptomyces* sp. P-56.

## Data Availability

Crystallographic data (excluding structure factors) for the obtained structures were deposited to the Cambridge Crystallographic Data Centre (CCDC) as supplementary publication No 1893406 CCDC. A copy of the data can be obtained on application to the CCDC (deposit@ccdc.cam.ac.uk).

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
