# Peer review of "Strain Streptomyces sp. P-56 Produces Nonactin and Possesses Insecticidal, Acaricidal, Antimicrobial and Plant Growth-Promoting Traits"

_microorganisms, 2023, doi:10.3390/microorganisms11030764_

Round 1

Reviewer 1 Report

The aim of this manuscript was to characterize the strain Streptomyces sp. P-56 to identify its active insecticidal metabolite and to investigate its other beneficial traits for plants. However, there are sever questions that might be the reviewer’ attention.

1.      The description of data in the abstract section were ambiguous.

2.      The novelty of this study was very ambiguous. Why done this work? There were no clear thoughts after reading the introduction.

3.      Table 2 should be present at the suitable location.

4.      The method section should be concise, it’s so long.

5.      The results section was too brief.

6.      The discussion section likes the introduction. There were few discussions that compared the data with other study.

7.      Biotests with plants should provide some pictures.

Author Response

The authors are very grateful to Reviewer for his valuable comments and suggestions. We hope that all comments have been addressed and the manuscript has been significantly improved. Responses to the Reviewer’s comments are marked with blue text. 

  1. The description of data in the abstract section were ambiguous.

Response: For concreteness, we have added the names of microorganisms to which the maximum antagonism of the P56 strain was shown (Lines 27-28).

  1. The novelty of this study was very ambiguous. Why done this work? There were no clear thoughts after reading the introduction.

Response: The aim of this study is described on lines 82-86. We tried to improve Introduction section to clarify the purpose of this work (lines 76-81).

  1. Table 2 should be present at the suitable location.

Response: Former Table 1 has been moved to line 397 and new number of this Table is 3. Former Table 2 (that you have mentioned) is now Table 1. In accordance with the rules of the Journal, the Table should be placed near to the place of its first citation. This table is cited on line 256 and placed accordingly on line 259.

  1. The method section should be concise, it’s so long.

Response: We have tried to describe the research methods in sufficient detail so that readers do not have questions and they can, if necessary, reproduce the experiments. Table 2 (new Table 1) and the related text have been removed from Methods section and inserted to the Results section on lines 389-406. We hope that such correction made this section more concise and shorter.

  1. The results section was too brief.

Response: The Results section has been complemented with Table 2 (new Table 1) and the related text (lines 389-406).

  1. The discussion section likes the introduction. There were few discussions that compared the data with other study.

Response: We tried to study in detail and cite the literature that is directly related to the issues under discussion. Now the list contains 60 references.

  1. Biotests with plants should provide some pictures.

Response: Since we did not find any visible negative effects on the development of plant seedlings, we did not consider it appropriate to take and present photographs. We decided describing this detail in the text only (lines 573-580).

Reviewer 2 Report

It is a very interesting work. The work is of particular importance from the point of view of introducing new ecologic solutions for protection against harmful organisms. The work is carefuly done and analytically confirmed. The manuscript is well organized and well written.

Author Response

Response: Dear reviewer! The authors are very grateful to you for the high appraisal of this work.

Reviewer 3 Report

This study is focused on the activity of nonactin obtained from the strain Streptomyces sp. P-56. The study is interesting and the objectives are clearly defined. The methodologies were correctly described and in my opinion allows to other researchers the replication of the experiments. The results and discussion are clear and clarify the readers on the main findings on the topic. There are some major and minor issues that should nevertheless be addressed.

Major limitation

- Figure 2. Another control with plants treated with ethanol (no extract) is mandatory. The observed effect can be cause (at least in part) by the extract solvent. This second control is necessary. Control plants treated with water can give you information on the quality of the plants but you need to test the effect of the solvents too.

Minor issues

- Some italics missing. E.g. Streptomyces in line 16.

- Table 1 . Please detail which solvents were used. Solvents A, B and C should be explained in the title or footnote.

- Figure 4 can be sent to supplements.

- Line 421. Please mention aversectin activity after Fitovern treatment to link text and table.

- table 5. Please add in the title or footnote what does “Nd”, “a, b” mean.

- Line 500. There is an error: Streptomycetes.

Author Response

The authors are very grateful to Reviewer for its valuable comments and suggestions. We hope that all comments have been addressed and the manuscript has been significantly improved. 

Major limitation

- Figure 2. Another control with plants treated with ethanol (no extract) is mandatory. The observed effect can be cause (at least in part) by the extract solvent. This second control is necessary. Control plants treated with water can give you information on the quality of the plants but you need to test the effect of the solvents too.

Response: In this experiment, the dried ethanol extract (DEE) of Streptomyces sp. P-56 was used. Ethanol was evaporated from the DEE. Please, see subsection 2.3 (Obtaining laboratory sample of the metabolic complex). So, the DEE contained no or just trace amount of ethanol. It contained some water, since the precipitate (DEE = “brown thick paste”) had a moisture content of 80%. In the experiment against peach aphid in the greenhouse with bell pepper (Figure 2), water suspension of 2 g L-1 DEE was used. This means that DEE was diluted with water 500 times and the possible presence of ethanol was negligible. Ethanol should be not toxic at such concentrations. Therefore we used treatment with water as a control.

Minor issues

- Some italics missing. E.g. Streptomyces in line 16.

Response: Correction has been done (line 16).

- Table 1 . Please detail which solvents were used. Solvents A, B and C should be explained in the title or footnote.

Response: Now this Table has number 3. Solvent’s names have been added to footnote of Table 3 (line 399).

- Figure 4 can be sent to supplements.

Response: This manuscript does not have supplements, and it seems inconvenient to create supplement from one small figure. To simplify the perception of this Figure and to easily compare it with Figure 5, we combined Figures 4 and 5 into one Figure. This is now Figure 4. Necessary changes have been also made to the text (lines 447-449, 456-459).

- Line 421. Please mention aversectin activity after Fitovern treatment to link text and table.

Response: Correction has been done (lines 532-533 and Table 5).

- table 5. Please add in the title or footnote what does “Nd”, “a, b” mean.

Response: Correction has been added to the footnote of Table 5 (lines 541-542).

- Line 500. There is an error: Streptomycetes.

Response: Correction has been done (line 623).

Round 2

Reviewer 1 Report

This manuscript can be accept for publication.

Reviewer 3 Report

The manuscript has been improved.